# Approaches to Bio-Cultural Diversity in British Columbia

David Zandvliet [1,*] , Shannon Leddy [2], Cate Inver [3], Victor Elderton [1], Brittney Townrow [4] and Lori York [5]

1 Faculty of Education, Simon Fraser University, Burnaby, BC V5A 1S6, Canada
2 Education, Curriculum and Pedagogy Department, University of British Columbia, Vancouver, BC V6T 1Z4, Canada
3 Gender, Race, Sexuality, and Social Justice, University of British Columbia, Vancouver, BC V6T 1Z4, Canada
4 Independent School Association of British Columbia, Vancouver, BC V5Z 0G3, Canada
5 Vancouver School Board, Vancouver, BC V6J 5K8, Canada
* Correspondence: dbz@sfu.ca

**Abstract:** This paper highlights an action research project into some decolonizing practices for environmental learning in the Canadian context of the British Columbia (BC) curriculum through case study and focus groups with Indigenous educators. A key finding taken from these discussions is that educators must strive to learn *with* Indigenous people not *about* them as the process of decolonization is about acknowledging multiple knowledge and value systems. We hosted two consultations with groups of Indigenous knowledge holders in the process of our work. The first consisted of a purposeful sample of Indigenous educators and academics drawn from *Coast Salish* communities in and around urban regions in coastal British Columbia (N = 20), the second draws on a purposeful sample of rural Indigenous educators and academics drawn from the Okanagan valley and British Columbia Interior (N = 10). The vignettes we share focus on how educational policy can be infused with Indigenous knowledges and pedagogical perspectives and, respectively, how policy changes might inform instruction on climate change education and a variety of other environmental topics. The revised framework we produce will guide teachers in their educational planning and support the implementation of environmental learning in diverse subjects using the dual lenses of bio-cultural diversity and inquiry—teaching about the environment and Indigenous knowledges together as an organizing theme for all teaching and learning. Our work highlights key aspects of our research involving practicing teachers, academics, provincial Ministries of Education and Environment as well as the Canadian Commission for UNESCO. The research articulates several important bridge-building activities: one linking research and policy, one bridging theory with practice, and most importantly, one bridging the inherent synergies among Indigenous and environmental knowledges.

**Keywords:** environmental and sustainability education; Indigenous education; experiential education; land-based education; place-based education

## 1. Introduction

Bio-cultural diversity as defined by Maffi is described as the diversity of life in all its manifestations: biological, cultural, and linguistic—which are interrelated within a complex socioecological adaptive system [1]. Maffi further relates that this diversity of life is made up not only of the diversity of plants and animal species, habitat, and ecosystems, but also of the diversity of human cultures and languages. While positive correlations have been described between biological diversity and linguistic diversity, our research focuses more centrally on social factors such as educational/cultural practices, as these have also been found to uniquely influence bio-cultural diversity.

This notion of bio-cultural diversity is akin to the Indigenous concept of land as first teacher. Positioning the land as a source of knowledge brings into focus the importance of a relationship. As Michell has phrased it [2], "we are the land, and the land is part of us. We are the context . . . When one aspect of nature is out of balance, all forms of life are affected"

(p. 17). Beyond considerations of diversity and inter-relatedness, this concept of land as teacher also implies responsibility. To acknowledge that we belong to the land means we also have a duty to maintain good relations with it. For Styres [3], "when land informs reflective practice, pedagogy, and storying, everything starts with and returns to the land, self is not/cannot be set apart from the interconnected and interdependent relationships embodied in land" (p. 718).

It is important to note that the concept of bio-cultural diversity is itself dynamic in nature and takes the local values and practices of different cultural groups as its starting point for sustainable living. For educators, the issue is not only to work to preserve/restore important practices and values, but also to modify, adapt, and create diversity in ways that resonate with both urban and rural communities. In this research, bio-cultural diversity is conceived as a reflexive and sensitizing concept that can be used to assess the different values and knowledge of all people—as a reflection on how we live now and in the future with biodiversity. As such, the concept must also be closely tied to climate and to issues of teaching and learning.

Our research also gives explicit attention to issues of power and privilege (using a critical pedagogy stance) as these are important mediating factors within the education milieu—particularly with reference to government mandated curriculums and teacher training practices. As Donald [4] frames it, "the origin of current human struggles to balance the desire for economic development with ecological sustainability derives from a deep forgetting of . . . simple truths" (p. 104). Our research aims to deeply describe the simple truth of our interrelatedness and interdependence with nature. As such, this work aims to provide a bridge linking ideas about Indigenous knowledge and pedagogy with western knowledge to better inform contemporary global society about how we may live now, and in the future, more sustainably. This focus leads to the key research questions which we hoped to address in our work:

(1) Are Indigenous knowledges and perspectives reflected adequately in the current framing of 'environmental learning' in the BC curriculum?
(2) Are there inherent synergies that emerge among Indigenous and Western perspectives on environment, climate, or other sustainability-related topics?
(3) What insights do Indigenous educators have about current attempts to reframe 'environmental learning' and assist BC teachers in this work?

*Rationale for the Study*

Our research with in-service teachers across the province of (so-called) *British Columbia* and Indigenous teachers and scholars in *Coast Salish* and *Syilx* (*Okanagan*) territories is pointing to the development of a revised framework that centers Indigenous knowledge and perspectives, while embracing the full diversity of perspectives, human and more-than-human, rooted in local contexts that connect with regional and global spheres. This paradigm shift has many precedents in the literature. In a chapter on environmental justice in the *International Handbook of Research on Environmental Education* for example, Randolph Haluza-Delay cites evidence of environmental racism in the United States and argues that environmental justice teachings need to be central to environmental learning going forward [5]. This claim builds on the work of the environmental justice movement in the U.S. and elsewhere, which brings attention to the ways environmental education has systemically centered the privileged in society and has not attended to the ways that social issues germane to environmental education pose disproportionate environmental harms for racialized and vulnerable communities [6].

Meanwhile, Indigenous territorial defenders and land-based practitioners have long demonstrated that there are clear links between the denial of Indigenous rights and environmental harm [7]. Learning from the land in Indigenous philosophies is an ongoing and vital experience of survivance, the resistance to colonization and revitalizing cultural, ancestral knowledge, and lifeways [8,9].

In Dolores Calderon's critique of social studies curriculum in the U.S. for example [8], there is a profound critique of the assumptions about what place means that even critical place-based education may perpetuate. For Calderon, the work of place-based learning needs to examine the histories of settler colonial land dispossession at the outset of understanding relationship to place as well as social positionality. Calderon insists that decolonizing the relationship to land is core to environmental education. This means "uncovering how settler colonial projects are maintained and reproduced, with understandings of land being one of the primary ways such identities are formed" (p. 28). She continues, "Without such exercises in decolonization, it is impossible to achieve goals of sustainability and the wedded notion of a community building that rejects anthropocentric and Eurocentric understandings of land and citizenship" (p. 28). There is a clear need for environmental education scholarship and practice to address these longstanding omissions, assumptions, and legacies of white supremacy and settler colonialism.

## 2. Materials and Methods

For our action research, it was recognized that environmental and climate education should consider at once multiple models for teaching and learning, as well as teachers' own pedagogical content knowledge to form a unique blend of interdisciplinary knowledge about specific learning contexts [10]. In this collaborative effort, we honour the diverse voices and methods that inform learning across British Columbia—while tapping into national and international academic discourses which inform the broader field.

The first part of our methodology (reported elsewhere) involved a sequential analysis of international education frameworks across several jurisdictions (e.g., Australia, New Zealand, the United States, and Canada). The second involves a community-based form of inquiry that has been termed participatory action research [11]. Here our research reconciles a global perspective with local knowledge by using the products of the first process as inputs into a second (community-based, participatory) process.

In the first stage of our action research, we reviewed and synthesized international environmental education perspectives comparable to the current framework in use in British Columbia. In this process, a critical focus was placed on the framework to inform the public consultation (second phase) of our research. Here, we proposed that the current *Environmental Learning and Experience* document [12] is inadequate and outdated (in terms of its background and practical information) to enable teachers to integrate Indigenous knowledges and climate change education effectively into the reformed BC curriculum. The purpose of this review was to look at place-based education in other parts of the world, specifically by including Indigenous perspectives on land-based learning, with the aim of providing positive and innovative ideas for the update of British Columbia's framework and to expose it to a sustained critique. This review also provided a global context in which to situate our work.

### 2.1. Participatory Action Research

Data from the first phase of research (comparative document analysis) was brought into a community-based form of inquiry in the second phase. In this type of research, scholars have generally described at least five approaches to participatory action research (or PAR) including: (1) action research in organizations, (2) participatory research in community development, (3) action research in schools, (4) farmer participatory research, and (5) participatory evaluation [11]. Conceptually, PAR originates from critical perspectives and practices that have been raised in the social sciences over the past four decades.

Traditional scientific approaches and educational practice often function, knowingly or not, to maintain specific hierarchical roles for researchers/subjects and teachers/students. PAR questions unequal power relationships inherent in these more traditionally run institutions (e.g., education or science) and instead, offers an approach to research that recognizes structural, institutional, and social injustices in our modern society which lead

to oppression. Importantly, the critical document review undertaken in the first phase of the proposed research informed a second participatory phase for the proposed research.

Key ideas emerging from the review included incorporating feedback and opinions through a consultation process and to clearly explain concepts of place-based learning or ecological literacy; to state guiding principles for teachers' practice; and to provide examples and resources that link to the new BC curriculum; providing flexibility for programs to reflect their community/place; and most importantly to centre local Indigenous worldviews, knowledges, and concerns in our revision work. These ideas about how to engage educators through workshops, information sessions, and community-based inquiry were brought into the second phase of the research process. This enabled a form of inquiry that places research capabilities into the hands of the "subjects" of the research, providing these individuals (in this case, Indigenous and environmental educators) with the research tools with which they can generate knowledge for themselves.

### 2.2. Consultative Methods

The focus and working groups conducted as part of this research occurred in a variety of communities around BC and included broad representation from various stakeholder groups, including the Ministry of Education, schools, informal education organizations, First Nations, university students, and academics. The structure of these meetings was congruent with the PAR approach in that they were co-lead and co-organized by community members and participants—with researchers acting as resources (alongside teachers, administrators, and officials) for the working part of the meetings. In the case of our consultations with Indigenous knowledge holders, this also applied but these consultations were held separately from other groups and ahead of our other consultations to foreground the variety of Indigenous perspectives in our work. Further, the transcripts and comments made by Indigenous colleagues during each of the two sessions we held were collated and transcribed by these educators themselves and member-checks were performed to ensure we were not misrepresenting the major points brought forward in each consultation.

The first consultation with Indigenous educators (February 2022) was hosted at the Van Dusen Gardens, a curated botanical garden in the Metro Vancouver region in the traditional Coast Salish territories of the *Musqueam*, *Squamish* and *Tsleil-waututh* First Nations. Twenty participants were drawn in a purposeful sample including educators and academics from each of these nations. Selected individuals were invited to a half-day meeting also attended by members of our research team. Lunch was provided as well as the opportunity to take a walk and be inspired by this tranquil urban garden setting.

The second consultation with Indigenous educators (October 2022) was hosted at the *En'owkin* Centre in *Sy'lix* First Nations' territory located in the BC Interior. Similarly, ten participants were included as a purposeful sample including a selection of elders, educators, and academics from across the region. Selected individuals were invited to a half-day meeting attended by members of our research team. Dinner was provided as well as the opportunity to take a walk in the natural bush setting of the semi-arid Okanagan valley.

For each of these sessions, participants were provided with the original government document: *Environmental Learning and Experience* [12], as well as a variety of other resources. Participants were organized into small working groups each tasked with re-visioning or re-purposing certain aspects of the original document (acting as quasi-editors for example). After each working session groups reported back on their work to the whole community (or to individual facilitators) to have their ideas further unpacked or enhanced. Researchers, teachers, and graduate students acted as resource persons and record keepers throughout this community-based process of data collection.

As a further enhancement to this process, community members were invited to make further presentations to the community about their land-based or place-based practices in environmental learning and were encouraged to comment on how our project should be communicated to the wider teacher audience and as to what format our work should take. A teacher/educator survey was also co-constructed with the Ministry of Education

and teacher colleagues to gain further data about teachers' needs in these areas. These communications typically continue (online) for several months after each face-to-face consultation, thereby extending the content re-visioning process in each community.

However, for the first part of our study, our focus was on Indigenous perspectives only—as this critical lens had been neglected in earlier revisioning attempts and we wanted to prioritize and (decolonize) the language and ideas used in the original Ministry of Education framework. In this, two extended focus groups were conducted with Indigenous educators. The first of these was rural in nature, involving educators drawn from the Okanagan and central BC interior; the second was urban in nature and involved educators drawn from the larger population centres of coastal British Columbia. It is important to note here that the findings from our Indigenous consultations differed substantially from findings that were drawn from other stakeholder groups; this was to be expected as we had determined in our research design that Indigenous voices had been underrepresented in the previous documents and that this was a problem we wished to correct. The reliability of this part of our research was ensured by repeated interviews and its validity was ensured by the long-term nature of our participatory research process which involved direct contact with respondents, co-constructing an accurate description of events, and preparing authentic transcripts of respondents' statements and contextual dialogue. The results of our consultations are presented in the results section.

## 3. Results

We share our qualitative results here as two interconnected vignettes, first in an effort to draw the reader into each of these thematic conversations and, further, to honor all of the contributions made during the talking circles we participated in to gather these results. Each vignette is drawn from an edited transcript, taken from a representative session with Indigenous educators and knowledge holders. We include with each a summary and a detailed transcript of the conversation co-constructed by Indigenous participants and our research team. Transcripts are thereby presented in their entirety in keeping with the principles of Indigenous data sovereignty and to share the full context of the conversation.

### 3.1. Vignette One: A Conversation on the Coast

**Facilitator:** (This will be a) curriculum framework document (so using) learning outcomes is (going to be) more difficult to do because the curriculum is on a renewal cycle, (it is a) political document. The framework (seeks) to look at curriculum and interpret it with this lens. Do we want this approach?

**R:** (The) topics that are suggested, stewardship, shared responsibility, seven generations—(these are) good concepts . . .

**M:** (there is a) land-based connection with our territories, (this) needs to be infused with the right language. Teachers being able to apply this to the classroom to learn about Indigenous territories in place . . .

**J:** What does it mean to connect with land as relative or land as teacher? (we need a) shift in the framework . . . (the) learning is cyclical . . .

**R:** Inquiry—as (an) educator (this is) infused in teacher education across subject areas (with) pathways for student agency, autonomy, choice, voice, (being) student-centered . . .

**J:** (The) inquiry lens could enable connection nature inquiry to storytelling, gardens, other pieces; weaving environmental conversation into every element of your classroom.

**R:** (Curriculum) relevance requires students' seeing themselves in the story . . . needing to see everything as story . . . how (do we) get students feeling that what we're learning is relevant to them?

**J:** Could you introduce two-eyed (seeing) as a way to connect the two frameworks as an introduction and then weave the 4 Rs all through the document-focus on those terms.

**Facilitator:** (There are some) epiphanies in this document, the framework document, (this) acronym wasn't made up, but is what it really was about . . . (CARE stands for Complexity, Aesthetics, Responsibility and Ethics in the current framework). Is that something to keep?

**J:** I like the acronym (but I am) struggling with all the internal terminologies (though) I like the word 'Connectedness' . . .

**R:** 'Ethics' seems like a word that comes from top-down and ascribes judgment. I would use 'respectful and responsible'

**J:** You have these terms (there) in the definition of ethics . . .

**R:** Are (these) the 4 Rs the ethics we're talking about?

**M:** (Yes) if you want to align it to the non-native world.

**G:** Using different words for the CARE acronym—4 Rs could be the R term . . .

**D:** Structure it with (the) two-eyed seeing approach, (we) could keep some of the existing document but add the parts that are missing and show how they're connected.

**R:** (It) would be a great exercise to situate the language teachers are familiar with and increasing their capacity for shifting their worldview . . . (this) considering how they dance together (we are) not asking teachers to totally throw out what they're doing but enhance what they're doing through a weaving practice, or braiding: (this is examining) two cultures with your own identity in the middle . . .

**G:** (Yes there is) perspective within that; depth perception needs more than one eye.

**R:** Have this document be inviting to teachers in shifting (their) perspective and (then) find ways into the beautiful . . . (the) cross-curricular idea is not reflective of the way Indigenous learnings are holistic . . . Let's have students represent their learning in ways that the students are drawn to (including) writing, voice, creation, music, math, architecture . . .

**M:** Our elders know that creativity is in everything we do; it's constant, moving, flowing (this is) where the students allowed to freely think and freely create (their) expression . . .

**R:** Acknowledge diverse ways of seeing. More than just Indigenous and Western. Have to open ourselves to thinking and learning in ways of all the diverse students in our classrooms are.

**Facilitator:** Environmental learning and Indigenous education are two places that share a vision of not being in a discipline, being holistic in (their) thinking . . . So what do we call this (document) when we're done? Land-based education? Place-based education?, Environmental learning? Indigenous and Western, thoughts about place? What do you call it that (is) to be inclusive of all those pieces?

**R:** It's all learning versus education . . .

**M:** What is environmental learning? So many things can pull into it. (We need to) challenge and think outside the box. We don't know all the answers. (For example) maybe you and your family know something, and you can share and give that . . .

**D:** Let people make connections and make them in your own places, in your own ways . . .

**R:** Our learners are not going to be experiencing learning all in the same ways . . . learners need to weave together their multiple ways of learning and questioning (and we need to) decolonize the roles of educators, away from transmission of knowledge frameworks (to) transformational connections of mind, heart and spirit (and) Problem based learning, students grappling through issues (we need to be) inviting them to think through what they need to learn in order to be able to do something about a problem . . . what world are we expecting to be able to have them be part of (with) the healing that is necessary?

**Facilitator:** One of the critiques over the years, (and the) document makes the case at the beginning—is that we don't need to plead with teachers and make the case anymore. (Instead) we need to help them do the work. Could this document become a 'guided inquiry', where each person comes through it with their own conclusions—based on their context and perspectives . . .

**J:** That (then) allows them to put themselves in that position and into their learning.

**R:** (Yes) a series of entry points and invitations . . .

**D:** (The) story piece could be really evocative . . . little story pieces/tasks/activities to illustrate

In summarizing the findings from this first, urban based conversation, we note that:

(a)    There is a need to help teachers shift away from old ways: transmission-based education to a more transformative and holistic learning—integrating mind, spirit, body, and heart.
(b)    There is a need to shift the language we use—to invite teachers into a sense of facilitating learning, while weaving their identities together with Indigenous and western perspectives.
(c)    Two-eyed seeing emerged as a useful organizing idea, while highlighting the need to include diverse learners.
(d)    It was stressed that experiential learning is cyclical and that understanding comes from holistic inquiry.
(e)    Finally, a key idea emerging is that we may need to structure the final framework as a guided inquiry for teachers with multiple entry points/outcomes.

*3.2. Vignette Two: A Conversation in the Interior*

**Facilitator:** Is there something worth keeping and building on (in this framework)? What's missing? Indigenous knowledges and pedagogy for example. What else? Beyond a document? What should we be pestering the Ministry of Education (about)?

**B:** (I am) drawn to the eco-cultural aspect of the document. Not just one partnership, multiple sites, multiple Indigenous people. (The) diversity aspect caught my attention . . . (however the) language is outdated. It is not about content, (it is instead) *how do we teach?*—a framework . . . Do we want the Ministry of Education endorsement? Some people say just publish with UNESCO . . .

The First Nations Education Steering Committee (FNESC) is (an) organization that represents and works on behalf of First Nations in British Columbia. FNESC has a mandate to support First Nations students (up to Grade 12) and advance First Nations education in BC.

**Facilitator:** (So) how do we create the networks? Not just relying on BC Ministry of Education endorsement. What else?

**J:** The Ministry of Education document is an important inroad. The cultural aspect of knowledge, out on the land. That is a problem, overarched by the whole education system. We can change how we do things for adults. The

Ministry of Education . . . de-schooling the document, (we) started working with the de-schooling concept when looking to establish *En'owkin* . . .

What is our mandate? (It is) contained within a larger circle of what's wrong out there . . . (We need to) de-school what the public school did to Indigenous people. Unlearn what they have learned in the public school system. To treat the land better and the community better. Transforming education, re-learning, breaking down the way the school is subject driven. De-schooling society.

We have a contribution to make, in terms of this transformation. The bush school model is one that everyone can use. *Learning in the real world, learning as if nature mattered* (espoused by the Centre for Ecoliteracy). A whole school decided to transform. Good work like that can be put (into) a pedagogical framework. How should it be framed so that it is doable? . . .

**J:** (We need to be sharing projects like) the edible schoolyard, transforming the whole school, planting food for those who needed it. Experiential learning, trauma-healing (a shared example about migrant workers)

**Facilitator:** How does the framework connect to the core competencies? (a) skills curriculum, this is how you become a good person . . .

**J:** That's the framework for the *Syilx* bush school. What do we need for the *Syilx* people and land to become healthy. The land healing the *Syilx* people and the *Syilx* people healing the land.

**S:** (I am) reflecting on (our) 2021–2022 *Tmix$^w$* science inquiry. In discussion with teachers, (it) recognizes there are teachers who do want to make changes. Logistics. Enough time, budget. What will the Ministry do to support the experiential learning endorsed by the (revised) document? (There is an) ethics and protocol for land-based knowledge. I am thinking about partnerships with . . . people from the local area . . . (we need to) showcase what is already happening . . . (and) here's who I need to talk to . . .

**B:** We need to transform teacher education as well. (These points are) worth keeping in the document: cultural diversity, connection to place. (but, we need to) add specific examples like the bush school. Healthy cultural diversity and eco diversity—(this is) where we need to get to. (There is an) outcome-oriented aspect to it and teacher training needs to be transformed. (There) is a school "monster", (and) economic "monsters" as well . . .

**D:** Knowledge situated in place. Linking the past, present, and future. Using pedagogy that incorporates that way of understanding (7th Generation thinking). Engaging in being. And being okay with that. That is the learning that is taking place. Students want to know how they're going to be graded, overshadow the process of relating, and the strengthening of relating. Students have notions of what the "standard" is but not what it means to be a good human. Capacity-building in place is (also) really important.

**C:** (Look at our) constitutionally protected rights that were enshrined when Canada adopted the constitution: The truth has not been the truth, it's been denial, "whitewashed". The local placed-based knowledge is our history. (Instead) the settler perspective takes precedent. (We have a) responsibility to share the truth, our connection to the land, our emotional, spiritual connection to the land, as a priority. It's a right. It is political. We live it . . .

**Facilitator:** (Yes) It's a failure of our education system.

**A:** Nothing about us, without us. Personally, (I) learned more when out on the Land. (We need to) invite local knowledge keepers. Learning with all the senses. Engaging . . .

**C:** (Yes) spending time outside, energy, . . . (We need to) make those connections readily available. Recognize (that) everything is interconnected. Sustainable learning is the highest form of learning. How (is) students' learning contributing to the health of Mother Earth? We all have to breath the air, drink water, healthy food to eat. Sustainable knowledge. The urgency . . .

**Elder:** *It is the Syilx responsibility to respect life.*

Discussion continued during the meal . . .
Summarizing the findings from this rurally based conversation, we note that:

(a) Conversations about Indigenous knowledges recognize plurality and cultural diversity (see Figure 1).
(b) First Nations ideas and knowledges have been systematically suppressed by dominant western (hegemonic) perspectives.
(c) The support of UNESCO may be a key referent for our work noting earlier directives such as the UN Declaration of Indigenous Rights of Indigenous Peoples [13].
(d) The need to acknowledge Indigenous knowledges as equally important along with western science in framing what constitutes the knowledge base.
(e) Climate and ecojustice are discussed as topics that are imbedded- and need to be made more explicit in the framework.

Perhaps the most important takeaway from the second discussion was a final comment from one participant who stated the need to "learn *with* Indigenous people not *about* us, as if we're under a microscope". When taken together, both vignettes indicate emerging approaches to weaving together dominant Western/Eurocentric learning frameworks with more socio-political dimensions from simply marginalized (or colonized) perspectives. Decolonial, critical, democratic, place-based, and land-based teaching all offer an array of possibilities for the future of environmental learning practices. Along with these paradigms, there are other permutations such as a critical pedagogy of place [14], and contextually rooted approaches such as Métissage, or intercultural education that blends Indigenous and Western sciences and ecological knowledges [15]. Finally, two-eyed seeing, a widely cited teaching brought forward by Mi'kmaw Elder: Albert Marshall, states that our eyes can each be a lens carrying the strengths of Indigenous and western knowledge, respectively [16]. Some of these approaches inform our discussion in the following section.

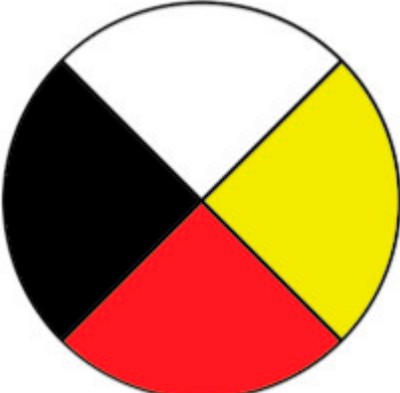

**Figure 1.** The iconic medicine wheel image is represented as a circle divided into four separately coloured quadrants. The number four has great significance in North American cultures. The black, white, red, and yellow medicine wheels frequently used in logos, on drums, and in art can be interpreted differently by individual communities (e.g., four directions, four seasons, etc.) and certain culturally significant animals, being part of the natural world, are also often included. The order of the colours is not the same in each culture, nor are the attributes [17]. For example, more than 45 versions of the Medicine wheel have been identified. This representation underscores the need to avoid sweeping generalizations about Indigenous cultures and to instead adopt a bio-cultural perspective recognizing the diversity of cultures and environments present on the land.

## 4. Discussion

Our research demonstrates that in the case of our first research question, Indigenous knowledges and perspectives are not reflected adequately in our current framing of 'environmental learning' in the BC curriculum and this will need to be addressed in our revisioning process. To achieve this we need to refer to the broader field of Indigenous education.

### 4.1. What Is Indigenous Education

Notably, the field of education on Indigenous knowledge has grown rapidly in the past three decades, initially led by scholar Gregory Cajete [18]. Following these developments, Lowan-Trudeau [19] outlined an 'Indigenous environmental education', noting that its emergence has paralleled a growing trend in North America and other parts of the world seeing a dramatic rise in "programs teaching Indigenous knowledge and philosophies for the benefit of both indigenous and nonindigenous students" (p. 404).

In discussing the role of story in Indigenous education, Styres [3] suggests it "is a discovery and creation of self in relationship; it is a process embedded in an examination of past experiences in relation to present and future actions". Not only does Indigenous education seek to address humans as holistic beings, but it also seeks to offer learning that is experiential and grounded in the needs of daily life. Indigenous education is concerned with learning to live in ways that are relational, ethical, and oriented towards collective well-being that includes both humans and all else that lives on and makes up the land.

Importantly, themes in science and environmental education in the US have also made strong links with Indigenous education [20]. One of the key points for our research was to consider the ways in which the process of decolonizing must be central to working with Indigenous knowledges. This means that we need to find ways to help non-Indigenous educators adopt an open stance to Indigenous ways of knowing that will allow them to truly hear what Indigenous educators have to offer. Notably, the work of Snively [21] offers examples of programs at the crossroads of Western/Indigenous approaches to education. Snively [21] whose work blended these two approaches together made the claim:

> Cross-cultural (science) education is not merely throwing in an Aboriginal story, putting together a diorama of Aboriginal fishing methods, or even acknowledging the contributions Aboriginal peoples have made to medicine. Most importantly, cross-cultural science education is not an anti-Western science. Its purpose is not to silence voices, but to give voice to cultures not usually heard to recognize and celebrate all ideas and contributions. It is as concerned with how we teach as with what we teach . . . (p. 38)

As noted earlier, there are continued calls to incorporate First Nations perspectives in K-12 school curriculum, across disciplines and across grade levels. In Canada, this is (in part) due to Canada's ratification of the UN Declaration of the Rights of Indigenous Peoples, which states that educators have a professional obligation to ensure the rights of all Indigenous Peoples [13]. This process naturally starts with educators including First Nations perspectives into their teaching. In response, BC has enacted specific actionable policies [22] within the teacher qualification standards noting that:

> Educators respect and value the history of First Nations, Inuit and Métis in Canada and the impact of the past on the present and the future. Educators contribute toward truth, reconciliation and healing. Educators foster a deeper understanding of ways of knowing and being, histories and cultures of First Nations, Inuit and Métis . . . (p. 5)

Since the beginning of the movement, Cajete [18] stated that if the current emphasis on Western oriented curricula continued, the Indigenous way of life would be further eroded. He points to a need for balanced integration of both Indigenous and western forms of education. Cajete's work is valuable in reminding us that while modernity may bring innovation, it is often accompanied by a loss of community. In his view, Indigenous

culture should be viewed as a complex intertwining of community and place. While there are resources available for educators to include First Nations content or cultural practices, Bilton et al. [23] remind us that "perspectives are not in the content, but the process" (p. 88). In other words, understanding perspectives requires understanding how knowledge is constructed and for what purpose.

Another key concept that has emerged is that of *Two-Eyed Seeing*. Initially developed by Murdena and Marshall Albert, two *Mik'maw* Elders from the *Eskasoni* First Nation in Nova Scotia in the context of science education at Cape Breton University [16], "Two-Eyed Seeing is the gift of multiple perspectives treasured by many aboriginal peoples and explains that it refers to learning to see from one eye with the strengths of Indigenous knowledges and ways of knowing, and from the other eye with the strengths of Western knowledges and ways of knowing, and to using both these eyes together, for the benefit of all" (p. 335). What is crucial about the framework is the understanding that there is no replacing of one paradigm with another, but rather we must learn to see things together and build on the strengths of each way of knowing, particularly where we find they overlap.

This concept also known as *Etuaptmumk* (two-eyed seeing) is also based on the principle that Indigenous and Western scientific ways of knowing are equally valuable, achievable, and inform how we live in the world [16]. It originates from the territory of the Mi'kma'ki people who have the longest history in Canada of living with colonisers, providing their Elders with a unique understanding of Western perspectives. The Mi'kma'ki word *Etuaptmumk* means "the gift of multiple perspectives".

Connecting back to our final research questions, these consultations have indicated clearly that there are inherent synergies emerging among Indigenous and Western perspectives on the environment, climate, and other sustainability-related topics. This point should lead us to question and reconsider what is meant when we use the term 'environmental learning' in our framework and other related research outputs. Further, Indigenous educators have important insights to consider as we begin to reframe 'environmental learning' and provide BC teachers with the resources they will require to undertake the important work of infusing environment, climate, and sustainability into diverse curriculums.

*4.2. What Is Environmental Learning?*

Acknowledging now the important cultural aspect of learning (and of inquiry) that emerged during our consultations, it is key to be explicit that there is no longer one answer to this question. Many participants in our study expressed tension with the term 'environment' seeing it as an anthropomorphic term, whereas others rejected the term 'land-based' education. Nevertheless, there is a need to facilitate students' understandings of what constitutes responsible actions toward the environment and help students to act responsibly in their personal lives. These actions are influenced by belief systems and personal limitations (cultural and physical), so student actions can take many forms. As such, 'environmental learning' will look different at different grade levels and in different in cultural or physical settings (see Figure 2).

Environmental learning (or whatever other term emerges from this work) should be seen as a flexible and interdisciplinary undertaking. Our research suggests a number of principles for organizing and conceptualizing this type of learning, and these begin with a focus on holistic cultural and ecological connections with the environment and end with the development of a collective ecological understanding in our communities. On the way to these, we can focus on developing an environmental appreciation, as well as the importance of taking responsibility for the collective stewardship of the environment for all life; this may lead to an *ethos* in which we are all required to be a participant. In this framework, we might continue with this revised mnemonic and metaphor of CARE (e.g., *connection*, *appreciation*, *responsibility*, and *ethos*) which can be used to describe the various forms that environmental knowledge can take.

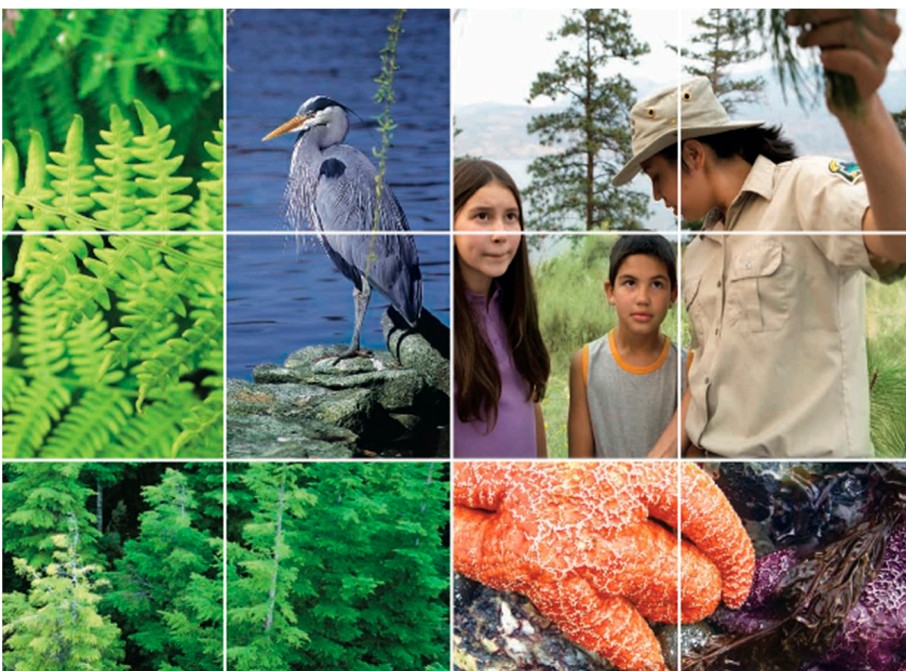

**Figure 2.** A cover image from the original *Environmental Learning and Experience* guide [12] first published in 2007. The mosaic of images represents a diversity of contexts for environmental learning in the province and makes explicit references to Indigenous knowledges. However, in the contemporary context the document is not seen as adequate in foregrounding Indigenous knowledge or Climate Action in teachers' praxis. This document was a key provocation in the current round of consultations with Indigenous knowledge holders and environmental educators.

Taking *CARE* with the environment, demonstrates the interdisciplinary nature of culturally or environmentally embedded concepts, while also showing the development of ideas that can lead towards deeper engagement with bio-cultural or environmental learning in the diversity of forms it may take. It is important to note further that the ultimate outcome of this type of education would be for students to strengthen their values towards environmental and cultural diversity. While this is described as an environmental ethic in the education literature when we are talking about individuals. However, as we are now considering environmental learning as a cultural undertaking, the collective ethos seems better suited to describe values that are imbedded in a community.

### 4.3. The Four R's

Finally, as we are discussing a collective form of ethics, it is important to note that when doing research or development activities within First Nations communities, there is an important set of ethics that have been articulated by leading First Nations scholars, Kirkness and Barnhardt [24], in doing this work. The guidelines are sometimes referred to as the *four R's*: signifying the concepts of *respect*, *relevance*, *reciprocity*, and *responsibility*. While these ideas were originally described in relationship to post-secondary environments, we will end our discussion here with a summary of the concepts as they relate to K-12 schools and what this means for our on-going action research.

The first R signifies respect for cultural integrity. A compelling problem that many First Nations students face when they go to school is a lack of respect, where 'school' represents an impersonal, intimidating and even hostile environment, in which little of what they bring in the way of cultural knowledge, traditions, and core values is recognized or respected. In this situation, FN students are expected to leave the cultural knowledge they hold and assume western cultural norms, a situation which is often substantially different from their own. In bringing students out of school and back 'into community', this creates an opportunity to honour this other knowledge base and empower students to

share their culture and knowledge for the benefit of all students. This is also a key tenant for the idea of bio-cultural diversity we are implementing in our work.

The second R signifies relevance to First Nations perspectives. If K-12 schools are to seriously respect the cultural integrity of First Nations' students and communities, they must also go beyond what has been described as literate knowledge and include the institutional legitimation of Indigenous knowledge and skills, which would include some aspects of oral traditions and ceremony. Kirkness and Barnhardt [24] put it this way: "Such a responsibility requires an institutional respect for Indigenous knowledge, as well as an ability to help students to appreciate and build upon their customary forms of consciousness and representation as they expand their understanding of the world in which they live . . . "

A third R signifies reciprocal relationships. One of the most troubling aspects of schooling for many First Nation students (and for all students for that matter) is the "dichotomy between the producers and the consumers of knowledge" in school settings. The institutionalized role of 'teacher' as the creator and dispenser of knowledge and the student as the passive recipient can prevent the establishment of the kinds of "human" relationships to which a diversity of learners will most likely respond to. Teachers and students engaging in 'reciprocity' are in a unique position to create a new kind of education, to engage in inquiry and to "formulate new paradigms or explanatory frameworks that help us establish a greater equilibrium and congruence between the literate view of the world and the reality we encounter when we step outside . . . " It is important to note that institutions (such as schools) can themselves engage in these reciprocal relationships thereby ensuring that learning also benefits the community where students reside [25].

The final R signifies responsibility towards action. In the context of First Nations perspectives on schooling, it is clear that education is not neutral: "It also means gaining access to power, authority, and an opportunity to exercise control over the affairs of everyday life . . . " For many students, this is a matter of necessity, because in order to survive the formal curriculum, they must also learn to navigate through the (sometimes) alien power structures of the school. Recent developments point towards an ongoing expansion of this thinking coming to be known as the six R's [26]. At a recent Indigenous knowledges and open education conference hosted by UBC, Kayla Lar-Son shared the 6 Rs of Indigenous Open Education Resources (OER), providing a framework for authors incorporating Indigenous knowledges into openly licensed teaching and learning materials. The six Rs provide an expanded framework for considering how we can work ethically to incorporate Indigenous knowledges in open educational resources and include the notion of relationships, including the concept of 'all our relations' within communities. Finally, the concept of reverence acknowledges that some knowledge is sacred and is not to be shared.

## 5. Conclusions

Our work remains in its early stages and so this paper has shared much more of our process for research—rather than a final product for educators to consider as our findings begin to unfold. Further, the idea of environmental learning (as we are now beginning to conceive it) is a cultural enterprise and will be influenced by our ideas about place, land, and community, as well as a unique context for authentic forms of inquiry. As such, the work may inform the current round of educational reforms—thus enabling the type of border crossing described by Giroux decades ago [27]:

> *Students must engage knowledge as a border-crosser, as a person moving in and out of borders constructed around coordinates of difference and power. These are not only physical borders, they're cultural borders—historically constructed and socially organized within maps of rules and regulations that limit and enable particular identities, individual capacities, and social forms . . . (p. 169)*

Getting back to the notion of bridge-building, our research has articulated several important bridge-building activities. First, because we are working with the Ministry of

Education as one of our research partners, our work aims to bridge research and policy in that our product will be a Ministry of Education endorsed guide for teachers. Second, because of our choice of an action research methodology, our work is actively building bridges between educational theory and teachers' practice—thereby empowering educators and Indigenous knowledge holders to share their ideas with the research community. Last, as our work highlights the inherent synergies among Indigenous and environmental knowledges, we provide a tentative bridge linking ideas about Indigenous ways of knowing and western thought processes around sustainability.

In conclusion, at the beginning of this paper we noted that the concept of bio-cultural diversity is dynamic in nature and takes the local values and practices of different cultural groups as its starting point for sustainable living. In our ongoing work with Indigenous knowledge holders, and environmental educators, the issue is not only to work to preserve and restore important cultural practices and values (for the benefit of all) but to modify, adapt, and create diversity in ways that resonate with both urban and rural school communities. In our continuing research, our intent is to highlight these approaches in detailed case studies (highlighting cultural differences among regions of the province), reflecting the various ways these frameworks can be applied in place-conscious ways. These case studies will be co-constructed with Indigenous educators and reflect the diversity of cultural practices in British Columbia.

Finally, the consultation process we use and the importance for considering local cultures in the interpretation of environmental learning is a key take away for this study and is important to consider for other regions or contexts both in Canada and internationally. In our research, we see bio-cultural diversity is a reflexive and sensitizing concept that can be used to assess the different values and knowledge of all people—as a reflection on how we live sustainably now and in the future. As we have described, these concepts are closely tied to those of culture, climate, and to issues related to teaching and learning about the environment.

**Author Contributions:** Conceptualization, D.Z. and S.L.; Investigation, D.Z.; Data curation, C.I. and V.E.; Writing—original draft, D.Z.; Writing—review & editing, S.L., C.I., V.E., B.T. and L.Y. All authors have read and agreed to the published version of the manuscript.

**Funding:** This research was funded by the Pacific Institute for Climate Solutions (PICS) under the *Opportunities Grant* funding category (Grant No. OP21DZ).

**Institutional Review Board Statement:** The study was approved by the Research Ethics Committee of Simon Fraser University and the Institutional Review Board at the University of British Columbia under Harmonized Review for research involving human subjects.

**Informed Consent Statement:** Informed consent was obtained from all subjects involved in the study.

**Data Availability Statement:** Data is contained within the article.

**Acknowledgments:** We would like to give thanks and acknowledge the many Indigenous scholars, educators and community members who participated in our consultations and graciously shared their knowledge and perspectives for this paper.

**Conflicts of Interest:** The authors declare no conflict of interest. The funders had no role in the design of the study; in the collection, analyses, or interpretation of data; in the writing of the manuscript; or in the decision to publish the results.

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
