# Peer review of "Approaches to Bio-Cultural Diversity in British Columbia"

_sustainability, doi:10.3390/su15086422_

Round 1
Reviewer 1 Report (Previous Reviewer 1)
To the authors,
I have seen regulations in paper about the current data noticeably as a summarized paragraphs as added sentences. In such a way, the information in this paper could be also summarized for the readers. Finally, I want to notice a point that this paper investigated the Canadian context of the British Columbia (BC) as a place. Therefore, I think that this situation as a pre-known place should emphasized in the title as a location for the readers.
Good luck.
Author Response
Thank you for your comments, we revised the title as follows:
Approaches to Bio-Cultural Diversity in British Columbia
Reviewer 2 Report (Previous Reviewer 2)
The paper deals with Approaches to Bio-Cultural Diversity in Education. The following comments should be addressed to improve the technical quality of the manuscript.
1. Can you provide more specific details in the Abstract on how the decolonization of environmental learning was approached in this project? Were there any specific actions or changes made to the BC curriculum, or was it more focused on theoretical discussions and recommendations?
2. While the dual lenses of bio-cultural diversity and inquiry seem like valuable frameworks for teaching and learning about the environment, how were these concepts specifically applied in practice by the participating educators? Were there any challenges or limitations in implementing these lenses?
3. The paper mentions that the project involved a variety of stakeholders, including practicing teachers, academics, and government organizations. Can you speak to the specific roles and contributions of these groups? Were there any power imbalances or conflicts that arose in these collaborations?
4. The paper acknowledges that the work is still in its early stages and that this paper primarily focuses on the research process rather than a final product. Given this, what are some specific next steps for the project, and how will these be approached?
5. While the paper highlights the importance of considering local cultures in environmental learning, it is not clear how this was specifically incorporated into the project. Can you provide more detail on how the project engaged with Indigenous knowledge holders and incorporated their perspectives and practices?
Author Response
Please see comments from previous revision:
Reviewer 2 (our responses in italics)
- Can you provide more specific details in the Abstract on how the decolonization of environmental learning was approached in this project? Were there any specific actions or changes made to the BC curriculum, or was it more focused on theoretical discussions and recommendations?
The following text was added to the abstract:
A key point taken from these discussions is that educators must strive to learn ‘with’ Indigenous people not ‘about’ them as the process of decolonization is about acknowledging multiple knowledge and value systems. The vignettes we share focus on how educational policy can be infused with Indigenous knowledges and pedagogical perspectives and respectively, how policy changes might inform instruction on climate change education and a variety of other environmental topics.
- While the dual lenses of bio-cultural diversity and inquiry seem like valuable frameworks for teaching and learning about the environment, how were these concepts specifically applied in practice by the participating educators? Were there any challenges or limitations in implementing these lenses?
These following comments have been added to the final comments in the conclusion section:
Our intent is to highlight these approaches in detailed case studies (from different regions of the province) reflecting the various ways these frameworks can be applied in place-conscious ways. These case studies will be co-constructed with Indigenous educators and groups in both the BC coastal region (Squamish First Nation) and the BC Interior (Sylix Nation). This work is ongoing and will be reported on later in our research.
- The paper mentions that the project involved a variety of stakeholders, including practicing teachers, academics, and government organizations. Can you speak to the specific roles and contributions of these groups? Were there any power imbalances or conflicts that arose in these collaborations?
The following texts were added to the manuscript to further describe the stakeholder consultations:
In the case of our consultations with Indigenous knowledge holders, this also applied but these consultations were held separately from other groups and ahead of our other consultations to ‘foreground’ the variety of Indigenous perspectives in our work. Further, the transcripts and comments made by Indigenous colleagues during each of the two sessions we held were collated and transcribed by these educators themselves and member-checks were performed in order to ensure we were not misrepresenting (or appropriating) the major points brought forth in each consultation.
It is important to note here that the findings from our Indigenous consultations differed substantially from findings that were drawn from other stakeholder groups, this was to be expected as we had determined in our research design that Indigenous voices had been underrepresented in the previous documents and that this was a problem we wished to correct. The results of these consultations are presented in the next section.
- The paper acknowledges that the work is still in its early stages and that this paper primarily focuses on the research process rather than a final product. Given this, what are some specific next steps for the project, and how will these be approached?
See comments for point number two (above)
- While the paper highlights the importance of considering local cultures in environmental learning, it is not clear how this was specifically incorporated into the project. Can you provide more detail on how the project engaged with Indigenous knowledge holders and incorporated their perspectives and practices?
Details about our consultation methods with the two Indigenous groups consulted have now been further elaborated on in the methods section Eg.:
We hosted two consultations with groups of Indigenous knowledge holders in the process of our work. The first consisted of a purposeful sample of Indigenous educators and academics drawn from Coast Salish communities in and around urban regions in coastal BC (N=20), the second draws on a purposeful sample of rural Indigenous educators and academics drawn from the Okanagan valley and BC Interior (N=10).
Reviewer 3 Report (Previous Reviewer 3)
Review Report
Article title: Approaches to Bio-Cultural Diversity in Education
I appreciate the effort the authors made in revising their original manuscript. The submitted manuscript clarifies almost all the points I raised earlier. The authors have improved the clarity of their writing and addressed most of my concerns.
Specific comments:
1. The text of the manuscript needs to be carefully proofread to avoid misprints, e.g., line 38 “[1}”, lines 121, 141, 412 and 528 “eg.” – “e.g.,” = exampli gratia (lat.), etc.
2. The source of Fig. 1 is not given in the figure caption. Since Figs. 1-3 are reproduced from other sources, you should have a written permission to re-publish them.
3. The text in Figure 1 is illegible. Either magnify the figure or redraw it.
4. The manuscript has relatively few literary sources to which it refers (only 27). In the Introduction section the current state of the research in the field should be reviewed and key publications should be cited.
In the text of the manuscript, it should be clearly stated that the reliability of the research was ensured by repeated interviews and the validity was ensured by the long-term nature of the research, direct contact with respondents, accurate description of events, and authentic quoting of respondents' statements.
Kind regards,
The Reviewer
Author Response
Dear Reviewer, thank you for your comments:
We corrected the grammatical errors noted as well as added the following text to the end of the methods section:
The reliability of this part of our research was ensured by repeated interviews and its validity was ensured by the long-term nature of our participatory research process which involved direct contact with respondents, co-constructing an accurate description of events, and preparing authentic transcripts of respondents' statements and contextual dialogue. Regarding the theoretical piece, we feel we have cited the appropriate literature regarding Indigenous knowledge. The field is new but we wanted to focus on the BC context in this paper. Thank you.This manuscript is a resubmission of an earlier submission. The following is a list of the peer review reports and author responses from that submission.
Round 1
Reviewer 1 Report
The information in this paper could be also summarized as bullets as clear and additional information but it could not. When I suggested this to authors, I had motioned as if I am a reader. However, I have to say that I am in same place as a thought.
I had thought that this paper could include a suggestion about bridge between environmental- indigenous learning and global education about how we would live now and in the future world in different culture and climate. This point would be a spark for future study. Unfortunately, I have not seen available a suggestion about this issue.
best wishes
Reviewer 2 Report
The paper deals with Approaches to Bio-Cultural Diversity in Education. The following comments should be addressed to improve the technical quality of the manuscript.
1. Can you provide more specific details in the Abstract on how the decolonization of environmental learning was approached in this project? Were there any specific actions or changes made to the BC curriculum, or was it more focused on theoretical discussions and recommendations?
2. While the dual lenses of bio-cultural diversity and inquiry seem like valuable frameworks for teaching and learning about the environment, how were these concepts specifically applied in practice by the participating educators? Were there any challenges or limitations in implementing these lenses?
3. The paper mentions that the project involved a variety of stakeholders, including practicing teachers, academics, and government organizations. Can you speak to the specific roles and contributions of these groups? Were there any power imbalances or conflicts that arose in these collaborations?
4. The paper acknowledges that the work is still in its early stages and that this paper primarily focuses on the research process rather than a final product. Given this, what are some specific next steps for the project, and how will these be approached?
5. While the paper highlights the importance of considering local cultures in environmental learning, it is not clear how this was specifically incorporated into the project. Can you provide more detail on how the project engaged with Indigenous knowledge holders and incorporated their perspectives and practices?
Reviewer 3 Report
Review Report
Article title: Approaches to Bio-Cultural Diversity in Education
The manuscript presents a case study that focuses on how educational policy can be infused with Indigenous knowledge and pedagogical perspectives. Furthermore, it studies how policy changes might inform instruction on climate change education and a variety of other environmental topics. The topic is interesting, however, the paper itself is not suitable for publication a scholarly journal in its current form. The comments are given below.
Specific comments:
1. The abstract should include a clear description of research sample and summary of the main research findings.
2. The introduction should include either clearly defined research hypotheses or research questions.
3. The authors report: “Our research aims to deeply describe the simple truth of our interrelatedness and interdependence.” The goal of the research is not clearly formulated because it is not measurable. What does it mean to describe deeply the simple truth of our interrelatedness and interdependence? Are you aiming for a complete description of some vaguely defined phenomena? In research paper, the research goal must be clearly formulated and achievable.
4. The authors shouldn’t report their findings at the beginning of the “Materials and Methods” section. It is the end of the “Introduction” section, where the main outcomes of the research should be outlined.
5. In the “Materials and methods” section, the subject of the research, the research sample (the method of its selection), the research methodology (the research methods used and their assignment to the relevant research hypotheses), the time schedule and the organization of the research should be presented in a concise manner. A precise description is missing.
6. The unstructured interview, which the authors used in their research, is one of the most demanding, the most advantageous, and relatively popular tools of qualitative research. On the contrary, the disadvantage of an unstructured interview is the problematic processing and interpretation of data, which was also reflected in the submitted manuscript. Although the authors provided a transcript of the relevant parts of the interview, they haven’t analyzed it. The unstructured interviews are evaluated mainly qualitatively, using the method of content analysis. The phenomenographic approach (a conceptual analysis) consists of the analysis of the obtained answers, their interpretation, and subsequent search or creation of a certain system, hierarchy, and categorization in the acquired set of knowledge/opinions. This procedure must be followed in the present paper.
7. How was the validity and reliability of the obtained data ensured?
8. In the “Discussion” section the authors should discuss the results and how they can be interpreted from the perspective of previous studies and of the working hypotheses. The authors should state here whether or not they have met the research objectives and to what extent.
9. The section ‘Summary’ needs to be better unfolded, highlighting what is new in the research and how the results of the study can be used in practice.
10. The paper should be supplemented with meaningful figures and results.